# The Anti-Cancer Effect of *Mangifera indica* L. Peel Extract is Associated to γH2AX-mediated Apoptosis in Colon Cancer Cells

**DOI:** 10.3390/antiox8100422

**Published:** 2019-09-22

**Authors:** Marianna Lauricella, Valentina Lo Galbo, Cesare Cernigliaro, Antonella Maggio, Antonio Palumbo Piccionello, Giuseppe Calvaruso, Daniela Carlisi, Sonia Emanuele, Michela Giuliano, Antonella D’Anneo

**Affiliations:** 1Department of Biomedicine, Neurosciences and Advanced Diagnostics (BIND), Institute of Biochemistry, University of Palermo, 90127 Palermo, Italy; marianna.lauricella@unipa.it (M.L.); cesare.cernigliaro@unipa.it (C.C.); daniela.carlisi@unipa.it (D.C.); sonia.emanuele@unipa.it (S.E.); 2Laboratory of Biochemistry, Department of Biological, Chemical and Pharmaceutical Sciences and Technologies (STEBICEF), University of Palermo, 90127 Palermo, Italy; valentina.logalbo@unipa.it (V.L.G.); giuseppe.calvaruso@unipa.it (G.C.); michela.giuliano@unipa.it (M.G.); 3Department of Biological, Chemical and Pharmaceutical Sciences and Technologies (STEBICEF), Section of Chemistry, University of Palermo, 90128 Palermo, Italy; antonella.maggio@unipa.it

**Keywords:** mango, apoptosis, reactive oxygen species, γH2AX, colon cancer cells

## Abstract

Ethanolic extracts from *Mangifera indica* L. have been proved to possess anti-tumor properties in many cancer systems. However, although most effects have been demonstrated with fruit pulp extract, the underlying molecular mechanisms of mango peel are still unclear. This study was designed to explore the effects of mango peel extract (MPE) on colon cancer cell lines. MPE affected cell viability and inhibited the colony formation trend of tumor cells, while no effects were observed in human dermal fibroblasts used as a non-cancerous cell line model. These events were a consequence of the induction of apoptosis associated to reactive oxygen species (ROS) production, activation of players of the oxidative response such as JNK and ERK1/2, and the increase in Nrf2 and manganese superoxide dismutase (MnSOD). Significantly, mango peel-activated stress triggered a DNA damage response evidenced by the precocious phosphorylation of histone 2AX (γH2AX), as well as phosphorylated Ataxia telangiectasia-mutated (ATM) kinase and p53 upregulation. Mango peel extract was also characterized, and HPLC/MS (High Performance Liquid Chromatography/Mass Spectrometry) analysis unveiled the presence of some phenolic compounds that could be responsible for the anti-cancer effects. Collectively, these findings point out the importance of the genotoxic stress signaling pathway mediated by γH2AX in targeting colon tumor cells to apoptosis.

## 1. Introduction

In recent years, many different findings have highlighted the beneficial value of food in the prevention and/or as a supportive strategy of the most common therapies for the treatment of many chronic diseases [1]. For these reasons, the term “functional food” is being used at an ever-increasing rate to indicate the antioxidant, anti-tumor or anti-inflammatory properties of many bioactive compounds extracted from plants or fruits [2,3,4].

In this scenario, particularly relevant seems to be mango (*Mangifera indica* L.), a crop belonging to the *Anacardiaceae* family that, according to historical records, has been cultivated in India and Southeast Asia for more than 4000 years [5]. Nowadays, mango orchards are spread around the world, in both tropical and subtropical environments, finding suitable areas for cultivation also in the Mediterranean area, where their spread first started in Spain, then, together with other tropical plants, also colonized Sicily, at least 20 years ago [6,7,8]. This fruit has received increasing interest from consumers who perceived its importance as a functional food rich in phytochemicals able to provide a healthy contribution to their diet.

Beyond the edible part of the fruit represented by the pulp (or mesocarp), the health-endorsing properties of mango have been also attributed to other different parts of the fruit such as the seed coat (endocarp), seed and peel (exocarp). In fact, although the pulp is endowed with a high polyphenolic and carotenoid content represented by mangiferin, gallic acid, gallotannins, quercetin, isoquercetin, ellagic acid, β-glucogallin and α- and β-carotene, many studies have supported the relevant composition of the other parts of mango fruit. In this regard, it has been observed that the peel and seed, usually discarded in fruit processing, are an important bio-source that can be exploited for their high content of polyphenols (mangiferin, quercetin, rhamnentin, ellagic acid and kaempferol), carotenoids, dietary fiber and vitamin E [9].

Several phytocompounds from different fractions of mango fruit have been shown to be powerful free radical scavengers, anti-microbial compounds, anti-inflammatory mediators or cancer preventive molecules [10], also exhibiting a strong cytotoxic activity towards many different human tumor models, such as blood [11], lung [12], breast [13], colon [14] and prostate cancer cells [15]. The anti-tumor action of mango extracts has been also demonstrated in vivo. In this regard, mango phenolic compounds showed a chemotherapeutic potential for suppressing tumor growth in breast cancer xenografts in mice, lowering the expression of a plethora of tumor associated proteins such as PI3K, AKT, hypoxia inducible factor-1α (HIF1α), and vascular endothelial growth factor (VEGF) [16].

In recent years, many findings aimed at eradicating cancer have been focused on those compounds that are endowed with a selective action, preferentially targeting tumors cells rather than normal ones. Some of these compounds have been shown to be involved in inducing genotoxic stress, oxidative injury and downregulation of protective and adaptive responses of tumor cells. From a therapeutic perspective, there are many examples of well-established cytotoxic antineoplastic agents commonly used for cancer treatment causing high levels of DNA damage by introducing double strand breaks (DSBs), regulating cell cycle checkpoints and leading to cell cycle arrest and/or activating cell death pathway [17,18]. When the genome of a cell is damaged or the action of genotoxins seriously affects the DNA stability, the cell detects these changes and attempts to repair the damage, recruiting DNA repair responses to avoid its transmission to daughter cells.

In this scenario DNA damage response activates a specific signaling pathway marked by the phosphorylation of histone 2AX (H2AX) on Ser139, a canonical key component able to monitor repair systems in DNA deeply affected by DBSs [19]. The phosphorylated form of H2AX (referred to as γH2AX) represents a histone variant considered as a ‘protagonist’ in different cellular scenarios and a reliable DNA DSBs biomarker [19]. To the best of our knowledge, γH2AX is the first marker that is upregulated by DNA damage response and accumulates at the site of the damage where it is visualized as foci, thus representing a sensitive tool to monitor cancer progression and efficacy of treatment [20].

With this in mind, our main focus was to explore the possible cytotoxic action of mango peel extract on colon cancer cells. Although it is already known in literature that other portions of this fruit, especially the endocarp, but also the pulp, are able to exert a cytotoxic action on tumor cells [14,21,22] a very limited number of investigations have explored the mechanisms underlying the anti-cancer properties of mango peel so far.

In addition, no data are available on the anti-tumor effect as well as on the phytochemical profile of mango grown in the Mediterranean area. In light of these considerations, this study was designed to investigate the anti-carcinogenic action of mango (*M. indica* L.) peel extract on colon cancer cell lines and evaluate its phytochemical composition using HPLC/MS (High Performance Liquid Chromatography/Mass Spectrometry) analysis.

Our study demonstrates for the first time that mango peel induces a selective mechanism of apoptotic cell death on colon cancer cells and that this effect is related to γH2AX-mediated genotoxic stress.

## 2. Materials and Methods

### 2.1. Preparation of Mango Peel Extract

Mango fruits were washed and the peel was removed, cut into small pieces and lyophilized. Afterwards, the small pieces were coarsely powered using a stainless steel grinder. The powder obtained was then solubilized in a solution of 50% ethanol in phosphate buffered saline (PBS) to a final concentration of 75 mg/mL and incubation was protracted overnight at 37 °C. Then, after centrifugation of the extract at 120× *g* for 10 min, the supernatant was recovered and subjected to a further centrifugation at 15,500× *g* for 10 min.

Hydro-alcoholic extract of mango peel (MPE) was stored at −20 °C until use. To perform treatments with a range of MPE concentrations, working dilutions were prepared in cell culture medium. For these experiments, cells were seeded, and after 24 h, when they covered the culture dish surface reaching a 70% confluence, MPE was added and incubation was protracted for the times reported in the results.

Final concentration of ethanol employed as vehicle had no discernible effects on colon cancer cells in comparison with control.

### 2.2. Reversed Phase HPLC/MS Experiments

Water, acetonitrile and formic acid HPLC/MS grade were used. HPLC samples were prepared by dissolving the obtained powder extract in MeOH (1 mg/1 mL). The employed HPLC system was an Agilent 1260 Infinity (Agilent Technologies Inc., Santa Clara, California, USA). A reversed-phase column [Phenomenex Luna C18(2) (150 mm × 4.6 mm, particle size 3 µm, Phenomenex Inc., Bologna, Italy) with a Phenomenex C18 security guard (4 mm × 3 mm), Phenomenex Inc., Bologna, Italy] was used. A 0.5 mL/min flowrate was set and column temperature was 30 °C. Injection volume was set at 25 µL. The used eluents were: phase A, 0.1% formic acid in water; phase B, 0.1% formic acid in acetonitrile. Employed gradient: 0–5 min, 5% B (isocratic); 5–15 min, from 5% to 15% B (linear gradient); 15–20 min, 15% B (isocratic); 20–25 min, from 15% to 30% B (linear gradient); 25–35 min, 30% B (isocratic); 35–45 min, 5% B (washing and reconditioning). MS total ion counts (TIC) was employed to monitor the eluate. Agilent 6540 UHD accurate-mass Q-TOF spectrometer (Agilent Technologies Inc., Santa Clara, California, USA) with a Dual AJS ESI source (Agilent Technologies Inc., Santa Clara, California, USA) was used to register mass spectra. All experiments were performed in negative mode. Desolvation gas was N_2_ (300 °C, 8 L/min); nebulizer (45 psig), sheat gas (400 °C, 12 L/min). The capillary potential was 2.6 kV and the fragmentor was 75 V. MS spectra range was 100–1000 *m*/*z*. Gallic acid and Mangiferin standards were supplied by Sigma-Aldrich (St. Louis, MO, USA).

### 2.3. Cell Cultures and Compounds

HT29, Caco-2 and HCT116 colon cancer cells (Interlab Cell Line Collection, ICLC, Genoa, Italy) and HDFa human fibroblasts (Gibco, Thermo Fisher Scientific, Monza, Italy) were cultivated in RPMI 1640 medium containing streptomycin (100 U/mL), penicillin (100 U/mL) and fetal bovine serum (10%) (Life Technologies, Milan, Italy), which was added after heat inactivation. Cultures were maintained at 37 °C in a 5% CO_2_ humidified incubator as previously reported [23]. All reagents and compounds, except where differently reported, were purchased from Sigma-Aldrich (Milan, Italy).

### 2.4. Cell Viability Assessment

Cell viability of different colon cancer cell lines (HT29 and Caco-2 adenocarcinoma cells and carcinoma HCT116 cells) and human dermal fibroblasts (HDFa) was determined by a colorimetric assay based on the metabolic use of 3-(4,5-dimethylthiazol-2-yl)-2,5-diphenyltetrazolium bromide (MTT) by viable cells. The analysis was performed as previously reported [8].

Morphologic changes of cells were observed using a Leica DMR inverted microscope (Leica Microsystems, Wetzlar, Germany) at 200× magnification. Pictures were acquired using a CCD camera and processed using IM50 Leica software (Leica Microsystems, Wetzlar, Germany).

### 2.5. Clonogenic Assay

For the clonogenic assay, HT29, Caco-2 and HCT116 colon cancer cells (200 cells/well) were seeded in 6-well plates and each condition was performed in triplicate. After 10 days, the cells were fixed in methanol, stained with 0.01% crystal violet for 40 min at room temperature. Finally, the plates were washed with water, air-dried and colonies containing more than 50 cells were counted manually. As reported by Wang et al. [24], the survival fraction (SF) was evaluated using the formula SF = number of counted colonies/number of plated cells × plating efficiency of the control group.

### 2.6. Analysis of DNA Damage and Cell Cycle Distribution

In order to determine changes in nuclear morphology the cells were stained with the bis-benzimide derivative Hoechst 33342, as suggested by Kelly [25]. For these studies, 8 × 10^3^ cells/well were seeded in a 96-well plate and before treatment were stained with Hoechst 33342 (2.5 μg/mL medium) for 30 min. Then, cells were washed with PBS, resuspended in culture medium and incubated with MPE.

Morphological changes induced by MPE treatment were examined using an inverted Leica fluorescent microscope (Leica Microsystems, Wetzlar, Germany) endowed with a 4′,6-diamidino-2-phenylindole dihydrochloride (DAPI) filter. Leica Q Fluoro software was used for image acquisition.

Cell cycle analysis by DNA content quantification was performed using flow cytometry on a Beckman Coulter Epics XL flow cytometer (Brea, CA, USA). For these analyses, cells were harvested, pelleted using centrifugation at 120× *g* for 10 min. Cells were washed once in cold PBS and centrifuged at 120× *g* for 10 min. Finally, 1 mL of propidium iodide hypotonic solution (50 μg/mL propidium iodide, 0.1% sodium citrate, 0.1% Nonidet P40 and 100 μg/mL RNase A) was added to cell pellets and incubation was protracted for 4 h in the dark at 4 °C prior to flow cytometry analysis. A total of 10,000 cells (events) for each sample were analyzed using FL3 channel (620 nm BP filter) to measure propidium iodide fluorescence. Analysis of data was performed using the Expo32 software.

### 2.7. Acridine Orange and Ethidium Bromide Double Staining for the Detection of Apoptosis

Apoptosis was detected using a dual staining with acridine orange and ethidium bromide (AO/EB) as reported by Ribble et al. [26]. Pictures were acquired using a fluorescent Leica microscope equipped with Rhodamine and fluorescein isothiocyanate (FITC) filters. Merge images were obtained combining pictures of both channels using Leica Q Fluoro software (Leica Microsystems, Wetzlar, Germany).

### 2.8. Assessment of Intracellular Generation of Reactive Oxygen Species

ROS generation was estimated using the fluorescent dye 5-(and-6)-carboxy-2′,7′-dichlorodihydrofluorescein diacetate (H2DCFDA). This compound passively enters into the cells where, after modification by intracellular esterases, it can be oxidized from intracellular ROS. Colon cancer cells (8 × 10^3^/well) were incubated with MPE at various times. Lastly, the medium was replaced and cells were incubated with 20 μM H2DCFDA at 37 °C for 30 min. Then, the medium was replaced with PBS supplemented with 5 mM glucose, and after 20 min, the fluorescence was directly visualized by means of a Leica fluorescence microscope (Leica Microsystems). Images were acquired using the Leica Q Fluoro software using a FITC filter. The fluorescence was measured using a Varian CARY Eclipse Fluorescence Spectrophotometer (Varian Medical Systems Italia SpA, Milan, Italy). Values were given in terms of mean fluorescence intensity.

### 2.9. Western Blotting Analysis

The analysis of proteins was carried out using western blotting. Briefly, after incubation with MPE, cells were lysed and protein extracts were prepared as previously reported [27]. 30 μg proteins/lane were resolved using SDS-PAGE and then electroblotted on a nitrocellulose membrane filter (Bio-Rad Laboratories Srl). The identification of proteins was determined using specific antibodies (1:200). Except for antibodies directed against caspase 9, PARP1 and ERK1/2 that were from Cell Signaling Technology (Cell Signaling Technology Inc., Beverly, MA, USA), all other analyses of proteins were carried out using primary antibodies distributed by Santa Cruz Biotechnology (Santa Cruz, CA, USA) and secondary antibodies from Amersham, GE Healthcare Life Science (Milan, Italy). All detections were performed using a ChemiDoc XRS System (Bio-Rad, Hercules, CA, USA) using Westar Ultra 2.0 enhanced chemiluminescence (ECL) reagent distributed by Cyanagen (Bologna, Italy). The intensity of the protein bands was quantified using Quantity One software (Bio-Rad) and normalized against a loading control protein that was not modified by the treatment and represented by γ-tubulin or β-actin (diluted 1:1000; Sigma-Aldrich, Milan, Italy).

### 2.10. Statistical Analysis

GraphPad Prism 5.0 software package (San Diego, CA, USA) was used to perform statistical analysis of data, which were shown as mean ± SD. The evaluation of significant differences between untreated and treated samples was performed using a Student’s t test. Differently, for the analysis of multiple groups, a one-way ANOVA test was applied. A *p* value < 0.05 was considered the threshold for statistical significance.

## 3. Results

### 3.1. HPLC-ESI-QTOF-MS Analysis Reveals the Composition of Mango Peel Extract (MPE)

HPLC-ESI-QTOF-MS analysis evidenced the presence of 16 polar compounds in MPE (see Table 1 and representative trace in Figure 1). The identified compounds can be grouped in organic acids, gallates and gallotannins, xanthones and benzophenone derivatives.

Three organic acids were identified: quinic acid (peak 2, at 6.70 min and *m*/*z* 191.0562), citric acid (peak 6, at 8.39 min and *m*/*z* 191.0198) and gallic acid (peak 7 at 8.63 min and *m*/*z* 169.0141) (Figure 1).

The presence of these compounds was in agreement with literature for mango peel [9]. Different compounds belonging to the family of gallates and gallotannins were also found: glucosyl gallate, (peak 3 at 7.00 min and 331.0673 *m*/*z*), digallic acid (peak 11 at 25.46 min and 321.0255 *m*/*z*), tetragalloyl glucose, (peak 13 at 26.31 min and 787.0998 *m*/*z*), methylgallate, (peak 14 at 26.36 min and 183.0301 *m*/*z*), pentagalloyl glucose, (peak 15 at 26.73 min 939.1100 *m*/*z*) and methyl-digallate ester, (peak 16 at 29.68 min and 335.0411 *m*/*z*). Peak 8, 11 and 12 were identified as three O-glucoside derivatives of benzophenone maclurin. Concerning xanthones, at 25.01 min and 421.0776 *m*/*z* (peak 9) mangiferin was found.

According to data previously reported in literature [9], the identification of galloylated benzophenone derivatives in mango peel provides evidence that galloylation of mangiferin and isomangiferin occurs before cyclization of benzophenones in the biosynthetic pathway. From a phytochemical point of view, the detection of galloylated maclurin could help to clarify the biogenesis of xanthone derivatives in mango.

In addition, lepidimoic acid was identified as peak 5 at 7.70 min with 965.2625 [3M − H]^−^
*m*/*z*. Lepidimoic acid was recently reported as a novel allelopathic substance, which has potent stimulating activity for the growth of other plant species [28]. To our knowledge, this is the first time that lepidimoic acid has been identified in mango fruits.

Table 1 also reports the quantification of free polar compounds in MPE, expressed as mg/100 g dry matter. Quantification of mangiferin and gallic acid was performed with the calibration curves of their own standards. According to previous studies, the gallates, gallotannins and maclurin derivatives were quantified using the calibration curve of gallic acid [29].

MPE presented free polar compounds of all the families. As usually observed for tropical fruits, phenolic acid derivatives, in particular the gallic acid derivatives, are abundant. Methyl esters of gallic and di-gallic acids were the major compounds with concentrations of 487.15 mg/100 g and 225.87 mg/100 g, respectively. Gallic acid derivatives have shown high antioxidant activity and health benefits ranging from neuroprotective action [30] to anti-cancer activity in many tumor systems [31].

### 3.2. MPE Induces Cytotoxic Effects and Morphological Changes in Colon Cancer Cells

To estimate the effects of MPE on different colon cancer cells, we performed cell viability tests using MTT colorimetric assay. For these studies, HT29, Caco-2 and HCT116, three different colon cancer cell lines, were screened at 24 h and 48 h after administration of different concentrations (15–600 μg/mL) of MPE. The analysis was performed in comparison to HDFa, primary human dermal fibroblasts isolated from adult skin, which was used as a non-tumor control, as previously reported [32]. As shown in Figure 2A, already after 24 h MPE treatment reduced cell viability of all the analyzed colon cancer cells in a dose-dependent manner. The effect started at 180 μg/mL MPE, and at 600 μg/mL concentration the evaluated residual viability amounted to 46%, 35% and 44% in HT29, Caco-2 and in HCT116 cells, respectively. This effect further increased by prolonging the incubation time of cells in the presence of MPE, reaching the maximum at 48 h of treatment with 360 μg/mL dose. In comparison, no significant cytotoxicity was observed when MPE was assayed on normal HDFa (−12%).

Having demonstrated that a consistent growth inhibitory effect occurs using 360 μg/mL MPE and that higher doses (600 μg/mL) induce unspecific cytotoxicity, all succeeding experiments were carried out using a 360 μg/mL dose.

As shown in Figure 2B reporting light microscope images, a large portion of colon cancer cells exposed to MPE treatment detached themselves from the substrate and exhibited clear morphological changes, such as cell shrinkage, roundness, and a vivid cell density reduction in comparison to the untreated cells. In contrast, these alterations were not visible in the normal control cell line HDFa, which appeared as a uniform array of cells abutting each other in both untreated and MPE treated conditions (Figure 2B).

The ability of MPE to inhibit cell growth was also confirmed using western blotting analysis of the proliferating cell nuclear antigen (PCNA, Figure 2C), a well-known marker of cell proliferation whose level was decreased by MPE treatment in all colon cancer cells.

### 3.3. MPE Treatment Inhibits Clonogenic Ability of Colon Cancer Cells

The inhibition of colon cancer cell proliferation was also confirmed by clonogenic assay (Figure 3), which measures the ability of single cells to proliferate, forming colonies containing at least 50 cells. The clonogenic potential is an important aspect of cancer behavior since it is strictly related to recurrence and metastatic ability.

Using different MPE doses (15–90 μg/mL), we observed that a concentration of 15 μg/mL MPE significantly reduced the clonogenic efficiency in Caco-2 (−36.4%) and HCT116 (−33.1%) cells, while modestly affected that of HT29 cells (−17.2%) (Figure 3) with respect to the clonogenic rate of untreated cells. The colony forming trend almost disappeared in both Caco-2 and HCT116 cells with 30 μg/mL, while such an effect occurred at a higher dose (90 μg/mL) in HT29 cells.

Taken together, these results highlight the ability of MPE to counteract the proliferative trend of colon cancer cells, reducing their viability and clonogenic potential.

### 3.4. MPE Triggers DNA Fragmentation and Apoptotic Cell Death in Colon Cancer Cells

Next, to investigate the nature of the cytotoxic effect of MPE we explored whether its action was accompanied with changes in the DNA integrity and activation of apoptotic cell death. As is evident from the cellular DNA content histograms reported in Figure 4A, MPE stimulation affected the cell cycle distribution of cells, promoting the accumulation of a remarkable percentage of cell population in the SubG0-G1 phase of the cell cycle with fragmented DNA, a hallmark of apoptotic cell death.

Such data was also corroborated by acridine orange/ethidium bromide double staining, which identifies morphological apoptosis-associated changes of cells [33]. Indeed, data provided evidence that MPE-treated cells are characterized by the presence of granular yellow-green nuclear staining (early apoptotic cells) or condensed orange nuclear staining (late-apoptotic cells), whereas a prevalence of diffuse green fluorescence was observed in untreated cells, indicating live cells (Appendix A).

The induction of apoptosis was also confirmed by the decrease in the level of both pro-caspase-9 and -3 as well as PARP1 breakdown, a known caspase-3 substrate (Figure 4B).

### 3.5. Assessment of Oxidative Stress Mediated Signaling in MPE-Treated Cells

It is well known that the DNA fragmentation can be a consequence of oxidative stress generation [34]. Thus, we estimated the ROS level by assessing cell ability to oxidize the fluorochrome H2DCFDA, a dye employed as a general indicator of cellular ROS. A time course analysis (Figure 5A) showed that the ROS level rapidly raised in MPE-treated cells, reaching the maximum at 30 min–1 h in all three colon cancer cells, when almost 90% of cells exhibited a pronounced green fluorescence detectable using either fluorescence microscopy or a fluorescence spectrophotometer. Thereafter, the effect was reduced, dropping to 35–40% of green fluorescent cells at 24 h of MPE exposure.

The addition of apocynin, a NADPH oxidase inhibitor, markedly reduced the ROS level at 1 h of treatment, thus suggesting the involvement of this multi-complex system in ROS generation. It is also interesting to note that the addition of N-acetylcysteine (NAC), a quencher of ROS, although it efficaciously rescued the early ROS production (30 min–2 h, Figure 5A), did not counteract MPE cytotoxicity observed after longer times of treatment (24–48 h, Figure 5B). These data strongly support the participation of ROS in MPE-mediated cytotoxic mechanism in the first phase of treatment.

The successive analysis evidenced that the observed events were accompanied by changes in cellular protein levels that are the main players of the cell response to stress. Firstly, we analyzed the phosphorylated extracellular signal-regulated protein kinases 1 and 2 (pERK1/2), mitogen-activated protein kinase family members that can be involved in cell proliferation, apoptosis and stress [35]. In accordance with the precocious ROS generation, pERK1/2 increase occurred in an early phase of incubation with MPE (1–8 h, Appendix A) and remained high up to 48 h (Figure 6), albeit it was counteracted by NAC addition only in the early phase of treatment (Appendix A). As reported in Figure 6, a remarkable increase in the level of pERK1/2 was observed after MPE exposure in all colon cancer cells analyzed.

The same treatment condition also promoted the upregulation of other MAP kinases involved in stress, such as phosphorylated JNK (pJNK), although its increase was less pronounced compared to that observed for pERK1/2.

In the next phase of our experiments, studies were carried out to ascertain whether cytoprotective events were activated in MPE-induced mechanism as a result of cell stress response. In this regard, the analyses were focused on two enzymes with antioxidant properties such as manganese superoxide dismutase (MnSOD), an oxidoreductase that removes the highly toxic radical species superoxide anion (O_2_•^−^), and catalase, an enzyme that detoxifies cells from hydrogen peroxide (H_2_O_2_). These two enzymatic activities represent important defensive systems that cells usually employ to detoxify themselves as a result of ROS overload [36]. As highlighted in Figure 7, in HT29 cells a considerable increase in the levels of MnSOD was determined compared to the untreated control cells after 48 h of treatment, while the levels of catalase remained substantially unchanged.

We also wondered whether the increase in MnSOD was associated with an increase in nuclear factor erythroid 2-related factor 2 (Nrf2), a transcription factor that, when phosphorylated and active, promotes the expression of antioxidant target enzymes such as MnSOD and catalase [37]. Western blotting analysis revealed the upregulation of Nrf2 upon MPE treatment for 48 h, an effect that was accompanied by the appearance of a band at a slower electrophoretic mobility corresponding to the phosphorylated form of the factor (Figure 7).

Taken together, these results seem to indicate that the early ROS generation, promoting both a widespread oxidative injury as well as pERK1/2 and pJNK upregulation, plays a role in the cytotoxic behavior of MPE.

### 3.6. DNA Fragmentation and Apoptotic Cell Death Induced by MPE is Related to γH2AX-Mediated Genotoxic Stress

To demonstrate that the effect of MPE is related to the induction of DNA damage, cells were stained with vital Hoechst 33342, a specific dye that allows the identification of DNA damage associated with chromatin condensation and fragmentation, as indicated by Kelly [25]. As can be seen in Figure 8A, untreated cells show a weak diffuse blue color given by the dye. On the contrary, when all three colon cancer cells were incubated with MPE the presence of an intense brilliant blue hue, an index of chromatin condensation and fragmented DNA, was observed.

In relation to the experimental evidence obtained, we turned our attention to the molecular events responsible for DNA damage, blockage of cell proliferation and oxidative stress injury. Firstly, we analyzed the phosphorylated form on serine 139 of H2AX histone, referred to as γH2AX. This protein, indeed, when activated by phosphorylation by the serine/threonine kinase ATM [38], represents an efficient system of recognition and repair of DNA double helix breaks [19]. Figure 8B shows that MPE treatment induced a consistent increase in γH2AX level as well as in the phosphorylated ATM protein (pATM) in all colon cancer cells, suggesting the recruitment of this repair system at DNA breaks. On the other hand, the involvement of the above-mentioned stress proteins as players correlated to γH2AX increase cannot be excluded. In fact, beyond pATM, H2AX phosphorylation could also be ascribed to other kinases such as pJNK [39], which in our experimental conditions was upregulated by MPE treatment (Figure 6), or the phosphorylated form of the kinase p38 [40].

Importantly, the DNA damage effect in treated Caco-2 and HCT116 cells was also accompanied by an increase in p53 level, the genome guardian protein capable of orchestrating a variety of DNA damage responses (DDR) and inducing apoptotic cell death [41]. In contrast, such an effect was not observed in HT29 cells, which are known to express a mutated form of this protein [42,43,44].

To explore a possible relationship between oxidative stress and DNA damage in MPE-treated cells, we further characterized the molecular mechanisms underlying MPE-mediated apoptosis in HT29. As depicted in Figure 9, the increase in γH2AX occurred precociously in stimulated cells appearing after 1 h and 8 h of incubation, thus suggesting the recruitment of this protein had already taken place in the first phase of treatment in concomitance with ROS generation. Indeed, the MPE-induced extent of γH2AX was conditioned by the addition of NAC, the quencher of ROS that, when co-administrated with MPE, completely suppressed the phosphorylation of H2AX in the first hour of exposure, an effect that disappeared for longer times of incubation (8 h).

## 4. Discussion

This paper aimed at analyzing mango peel extract for its ability to tackle the tumor behavior of different human colon cancer cells and explore its mode of action.

The choice of this analysis was sustained by the observation that many studies report the health endorsing properties of mango from tropical areas, while no data are currently available on peel from mango cultivated in Sicily (Balestrate, Italy), especially on its anti-cancer activity. Sicily is characterized by a particular pedoclimatic environment that seems to be favorable for mango cultivation conferring particular properties to the orchards. For this purpose, we were interested in characterizing the chemical composition of MPE obtained from mango grown in the rural Sicilian areas as well as to test its effects on tumor cell systems.

Significantly, our results provided evidence that MPE exhibits a selective anti-carcinogenic action against colon carcinoma cells inducing a γH2AX-mediated genotoxic and apoptotic effect, while it turned out to be ineffective on human dermal fibroblasts. MPE effect is accompanied by early ROS generation causing both the activation of phosphorylated stress proteins (as pJNK) as well as an extended DNA damage that committed colon cancer cells to γH2AX-mediated apoptotic demise. Overall, these data are in accordance with results demonstrating that the increase of pJNK could be also ascribed to the activation of an apoptotic cell death program, as recently shown in cells of chronic myeloid leukemia treated with resveratrol [45], one of the main polyphenols present in plants.

We also demonstrated that MPE promotes the upregulation of pERK1/2, members of the mitogen-activated protein kinase family capable of mediating cell proliferation, apoptosis [35] and inducing ROS-mediated NADPH oxidase [46]. Beyond the activation of the stress kinase pJNK and pERK1/2, the oxidative injury sparked by MPE treatment was also accompanied by the recruitment of Nrf2, a well-known repair system active in the defense and protection of cells from toxic and oxidizing events. Nrf2 is a transcription factor that, when phosphorylated [47,48], translocates into the nuclear environment where it transactivates the expression of a battery of genes with antioxidant action such as superoxide dismutase (MnSOD), catalase, NAD(P)H quinone reductase, glutathione S-transferase and HO-1 [37,49]. Our experiments have highlighted that the activation of protective responses such as Nrf2 and its target MnSOD probably attempt to counteract MPE-mediated oxidative injury.

Taken together, these data, supporting the participation of oxidative stress in the MPE effect, lead us not to exclude a possible involvement of NO and nitrosative stress in our systems, which we aim to investigate in future. Indeed, NO exhibited a dual action on many cancer models, enhancing cell permeability and retention of chemotherapeutics as well as killing tumor cells [50]. On the other hand, our chemical characterization of MPE using HPLC/MS indicated the presence of mangiferin, a well-known bioactive xanthone that has been proved to increase both NO production and eNOS expression in some cell systems [51]. Such an aspect could be of particular interest since some NO inducers or NO-donating compounds have been recently identified as new, promising anticancer therapeutics against blood and solid tumors [52,53,54,55].

In addition to the extended DNA damage promoted by MPE, as well as an increase in γH2AX level, an increase in that of pATM and p53 was also observed. As reported in the literature, γH2AX represents the first step for the recruitment of DNA repair proteins at the damaged sites. H2AX is one of the main five histones belonging to the H2A histone protein family, whose relevance has been highlighted for its phosphorylated form at ser139, referred to as γH2AX, and that represents a sensitive indicator of damaged DNA by environmental and exogenous insults [56]. These observations are in accordance with our data showing γH2AX increase related to the apoptotic cell death induced by mango peel. The level of γH2AX rapidly increased in the early phase of treatment and remained high along all the process that committed colon cancer cells to apoptotic demise. These data were in accordance with Rogakou [57] who, studying the kinetic of γH2AX formation in Jurkat cells exposed to the effective apoptotic inducer staurosporine, provided a detailed examination of γH2AX timing and DNA fragmentation of dying cells. Similar effects were also described in other apoptotic systems such as staurosporine-treated HL60 cells [58] and etoposide-treated Jurkat cells [57]. Solier et al. [59] also analyzed the presence of γH2AX and phosphorylated H2B in a novel cell entity named apoptotic ring, where the histone variants are recruited and activated by DNA damage response kinases. In line with this observation, more recently a novel apoptotic signaling pathway dependent on the activation of the H2AX-CARP1 (cell cycle and apoptosis regulatory protein) axis has been demonstrated as a key event in the transduction of apoptosis following DNA damage [60].

On the whole, the data reported here provide evidence that MPE stimulation triggered an apoptotic cell demise orchestrated by ROS-mediated genotoxic stress and a remarkable increase in γH2AX level. However, since these effects had already appeared in the first hours of incubation, when cells are still alive, it is conceivable that along the pathway that commits cells to death, tumor colon cells try to activate cytoprotective systems such as Nrf-2 and its transcriptional target MnSOD, which hopelessly fail in sustaining tumor survival. Thus, cells become disarmed in their defense systems and collapse by apoptosis. A schematic representation of MPE anticancer activity on colon cancer cells is reported in Figure 10.

Interestingly, the characterization of the polyphenolic profile of MPE using HPLC/MS provided evidence that different families such as organic acids, gallates and gallotannins, xanthones and benzophenone derivatives enrich this fruit and could be the main players in the analyzed mechanism. In addition, differently from other previous studies, we demonstrate for the first time the presence of lepidimoic acid, a pectic disaccharide, in mango peel. To our knowledge, this is the first evidence of this compound in mango.

## 5. Conclusions

Overall, our results correlate with data present in the literature on the anti-tumor action of mango peel extract. However, here we provide a new insight into the mechanism of action of MPE preferentially targeting colon cancer cells rather than normal ones. The chemical characterization of MPE sheds light on the composition of possible phytochemicals responsible for these selective properties. Moving forward, our future studies will aim to better understand the role of molecules contained in MPE. To this purpose, the different phytochemicals found in MPE will be tested alone or in combination in future investigations to explore possible synergistic interactions. In addition, a putative action of MPE in combination with the most common chemotherapeutics will be also considered with the aim to provide a new insight into the anti-tumor potential benefits of mango peel as a supportive strategy for antineoplastic therapies.

## Figures and Tables

**Figure 1 antioxidants-08-00422-f001:**
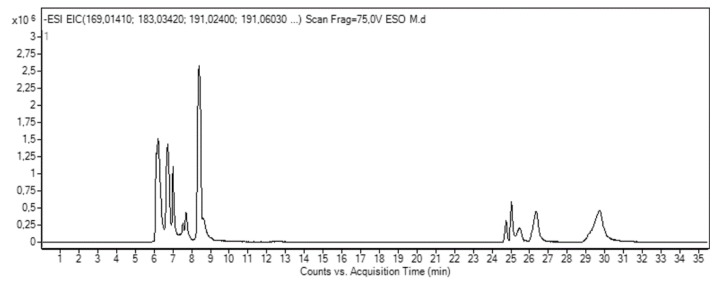
Representative high performance liquid chromatography-electrospray-ionization-quadrupole-time of flight (HPLC/ESI/Q-TOF) trace of mango peel extract.

**Figure 2 antioxidants-08-00422-f002:**
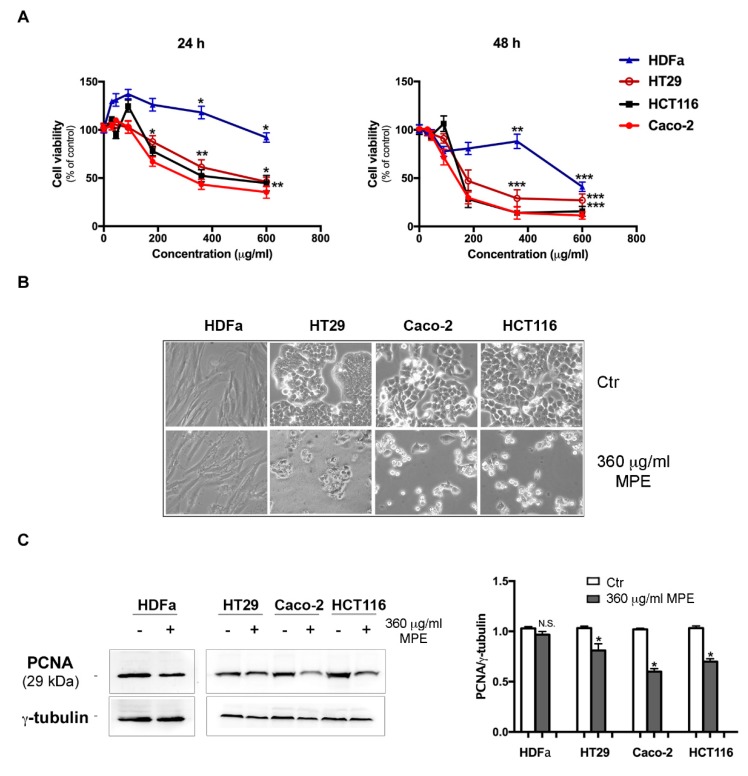
MPE effects on cell viability and the level of proliferation marker proliferating cell nuclear antigen (PCNA) in human dermal fibroblasts and colon cancer cell lines. (**A**) Human dermal fibroblasts (HDFa) and colon cancer cells (HT29, Caco-2 and HCT116) were incubated in the presence of different concentrations of MPE for 24 and 48 h as reported in the Materials and Methods section. Then, the percentage of viable cells was evaluated using 3-(4,5-dimethylthiazol-2-yl)-2,5-diphenyltetrazolium bromide (MTT) assay. The values reported in the line chart are the means of three independent experiments ± SD. Statistical significance was assessed using the Student’s t-test: (*) *p* < 0.05, (**) *p* < 0.01 and (***) *p* < 0.001 compared to the untreated sample. (**B**) Morphological changes induced by 360 μg/mL MPE were acquired after 48 h of treatment by an inverted Leica microscope. A picture taken at a 200× magnification by IM50 Leica software is reported in the figure. (**C**) Western blotting analysis of PCNA protein expression in HDFa and colon cancer cells. The correct protein loading was ascertained by immunoblotting for γ-tubulin. Densitometry values are averaged from three independent experiments normalized to γ-tubulin. (*) *p* < 0.05 compared to the untreated sample. N.S., not significant.

**Figure 3 antioxidants-08-00422-f003:**
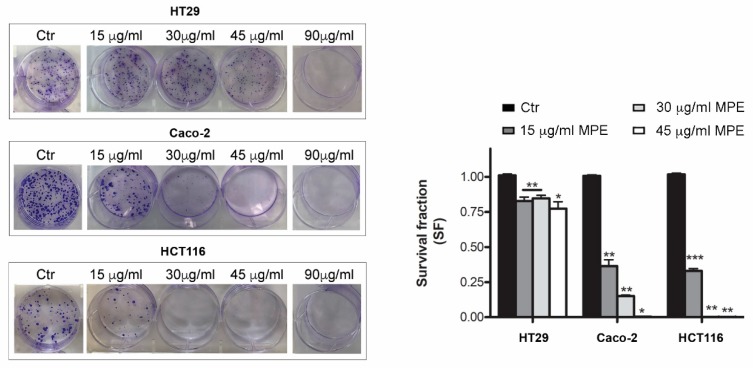
Cytotoxic effect of MPE evaluated using clonogenic assay in colon cancer cells. Culture dishes with crystal violet stained colonies treated with MPE. The ability of cells to produce colonies, as reported in Materials and Methods, was evaluated after 10 days. Pictures of a representative experiment are reported. In the right panel statistical results of colony-forming assays were reported as survival fraction (SF) of colonies number with respect to untreated condition used as control. Statistical significance was assessed by the Student’s t-test: (*) *p* < 0.05, (**) *p* < 0.01 and (***) *p* < 0.001 compared to the untreated sample. The data in the bar chart are expressed as mean number ± SD of three different experiments.

**Figure 4 antioxidants-08-00422-f004:**
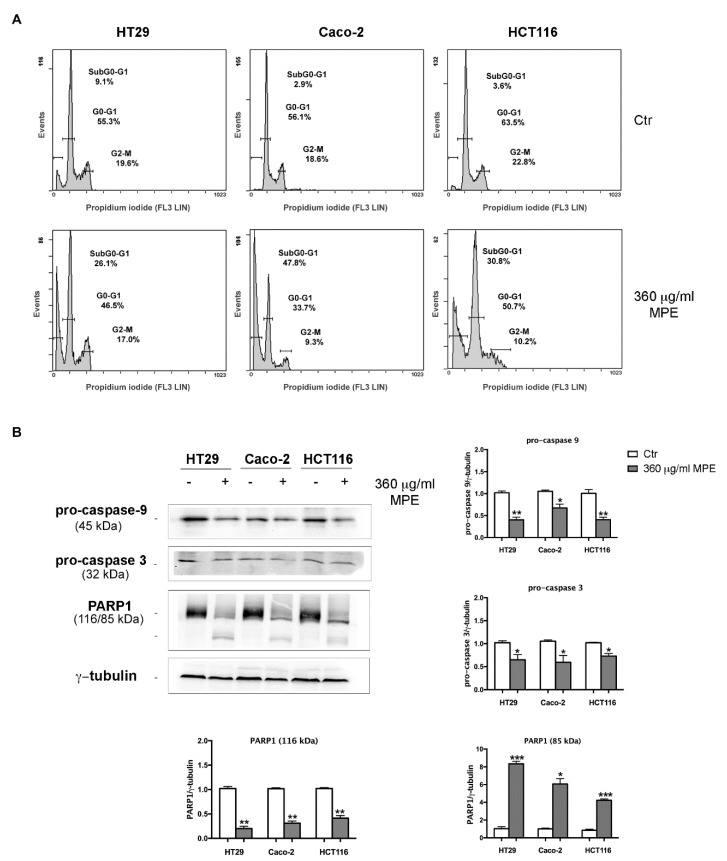
MPE effects on cell cycle phase distribution, caspases activation and PARP1 fragmentation. (**A**) Flow cytometry analysis of cell cycle phase distribution of HT29, Caco-2 and HCT116 cells exposed to MPE treatment for 48 h. (**B**) Western blotting analyses of pro-caspases-9 and -3 and PARP1 were performed as reported in Materials and Methods. The correct protein loading was ascertained by immunoblotting for γ-tubulin. Representative blots of three independent experiments and densitometry analysis histogram are reported. (*) *p* < 0.05, (**) *p* < 0.01 and (***) *p* < 0.001 compared to the untreated sample.

**Figure 5 antioxidants-08-00422-f005:**
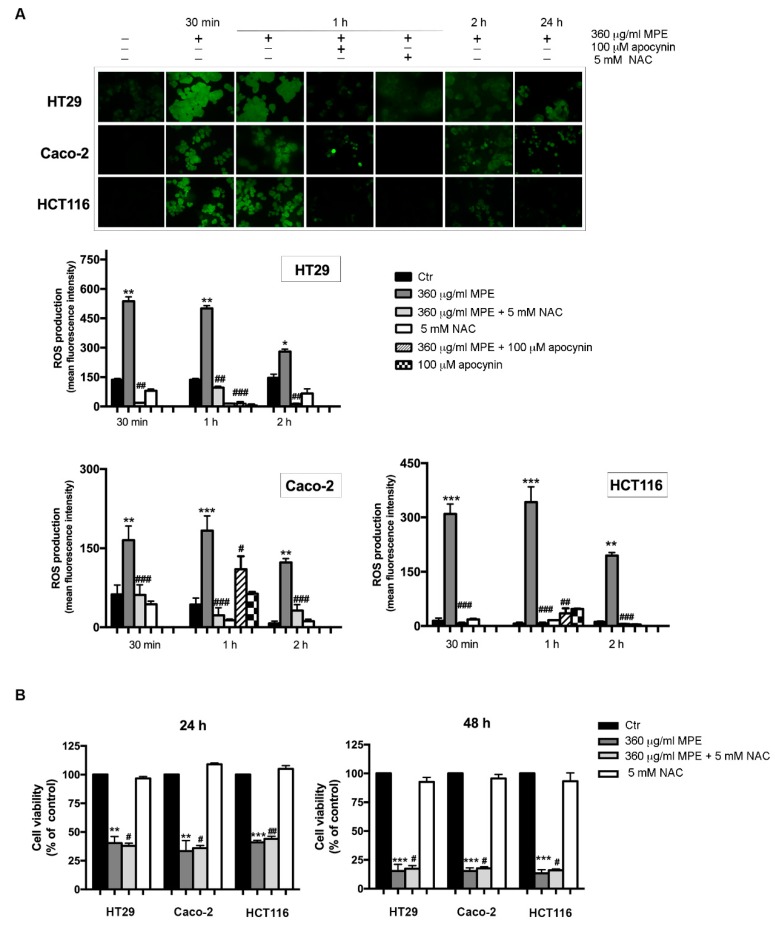
MPE treatment stimulates the intracellular reactive oxygen species (ROS) generation. (**A**) Micrographs of fluorescence microscopy showing ROS generation after treatment of colon cancer cells with MPE in the presence or a fluorescein isothiocyanate (FITC) filter (upper panel). The fluorescence was measured by a Varian fluorescence spectrophotometer and values were reported as mean fluorescence intensity (lower panel). (**B**) Cytotoxic effect of MPE on colon cancer cells treated with or without 5 mM NAC. Cell viability was assessed by MTT assay, as reported in Materials and Methods. The data are expressed as mean value ± SD. (*) *p* < 0.05, (**) *p* < 0.01 and (***) *p* < 0.001 compared to the untreated sample. (#) *p* < 0.05, (##) *p* < 0.01, (###) *p* < 0.001, compared to MPE-treated sample.

**Figure 6 antioxidants-08-00422-f006:**
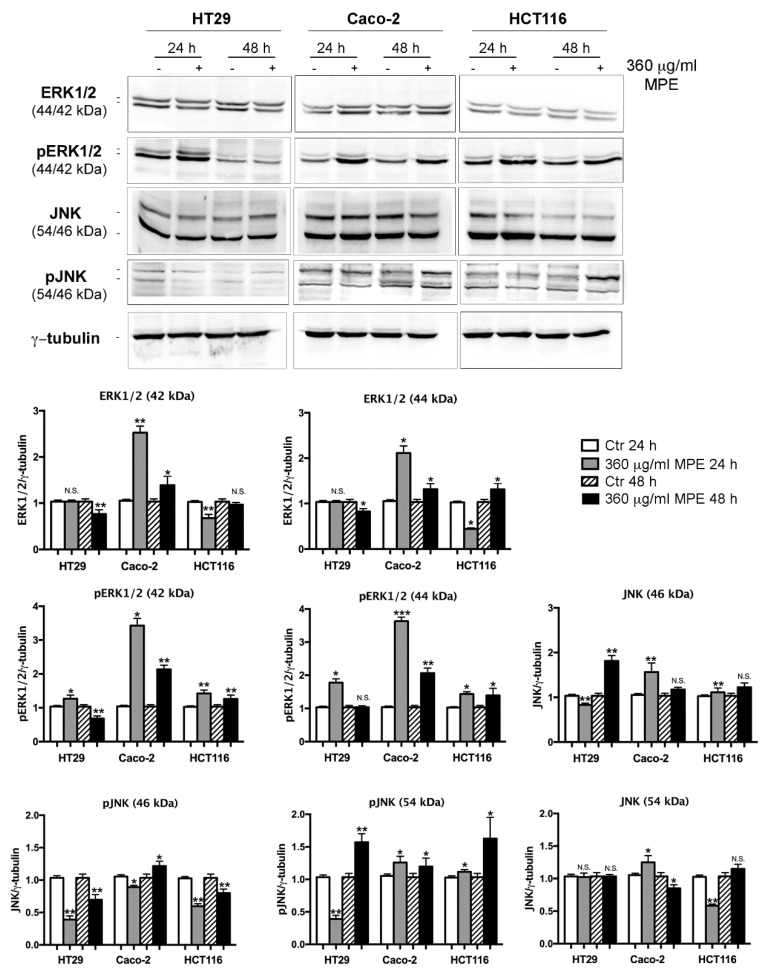
MPE treatment upregulates the main stress protein players pERK1/2 and pJNK. Colon cancer cells were treated with MPE for the indicated times, then Western blotting analyses of ERK1/2, pERK1/2, JNK and pJNK were performed, as reported in Materials and Methods section. Proteins were detected using specific antibodies. The correct protein loading was ascertained by immunoblotting for γ-tubulin. Representative blots of three independent experiments and densitometry analysis histograms are depicted. (*) *p* < 0.05, (**) *p* < 0.01 and (***) *p* < 0.001 compared to the untreated sample. N.S., not significant.

**Figure 7 antioxidants-08-00422-f007:**
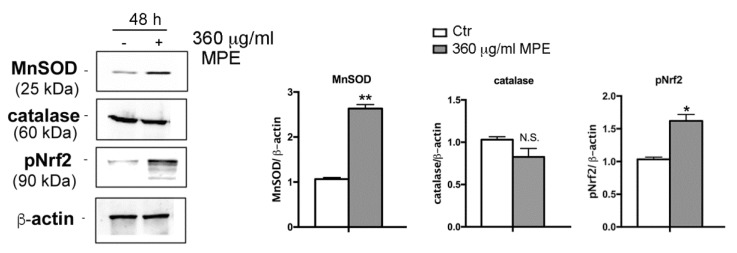
MPE treatment upregulates antioxidant responses. Colon carcinoma HT29 cells were exposed to MPE treatment for 48 h. Then, cells were lysated and proteins were analyzed using western blotting by using specific antibodies directed against MnSOD, catalase and Nrf2, as reported in Materials and Method section. The correct protein loading was ascertained by immunoblotting for β-actin. Representative blots of three independent experiments and densitometry analysis histograms are depicted. (*) *p* < 0.05 and (**) *p* < 0.01 compared to the untreated sample. N.S., not significant.

**Figure 8 antioxidants-08-00422-f008:**
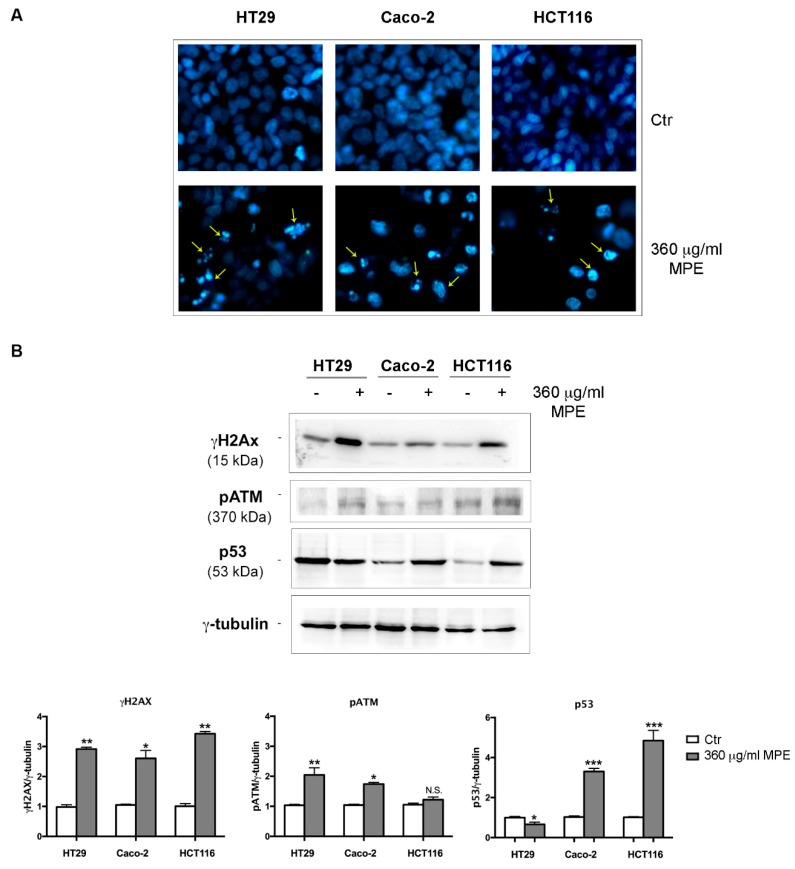
MPE effects on DNA damage markers. (**A**) The effect of MPE on DNA was evaluated using Hoechst 33342 staining in colon cancer cells incubated with MPE for 48 h. The pictures (original magnification 400×) were acquired using a 4′,6-diamidino-2-phenylindole dihydrochloride (DAPI) filter with a Leica fluorescent inverted microscope using Leica Q Fluoro software. Yellow arrows indicate condensed or fragmented chromatin. (**B**) MPE provoked the phosphorylation of both H2AX (γH2AX) and ATM (pATM) and p53 upregulation. All colon cancer cells (HT29, Caco-2 and HCT116) were treated for 48 h in the presence of 360 μg/mL MPE. Then, cell lysates were analyzed using western blotting with specific antibodies directed against the proteins of interest, as reported in Materials and Methods. The correct protein loading was ascertained by immunoblotting for γ-tubulin. Representative blots of three independent experiments and densitometry analysis histogram normalized to γ-tubulin are reported in the bottom panel. (*) *p* < 0.05, (**) *p* < 0.01 and (***) *p* < 0.001 compared to the untreated sample. N.S., not significant.

**Figure 9 antioxidants-08-00422-f009:**
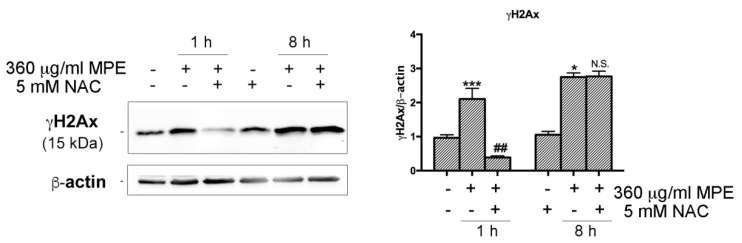
MPE promotes early activation of γH2AX via oxidative stress. HT29 cells were treated with MPE for the indicated times in the presence or absence of NAC. Lastly, western blotting analyses were performed in order to study the effects of the treatments on the total level of γH2AX. The correct protein loading was ascertained by immunoblotting for β-actin. The results are representative of three independent experiments and densitometry analysis histograms are reported normalized to β-actin. (*) *p* < 0.05 and (***) *p* < 0.001 compared to the untreated sample. (##) *p* < 0.01 compared to MPE-treated sample. N.S., not significant compared to MPE-treated cells.

**Figure 10 antioxidants-08-00422-f010:**
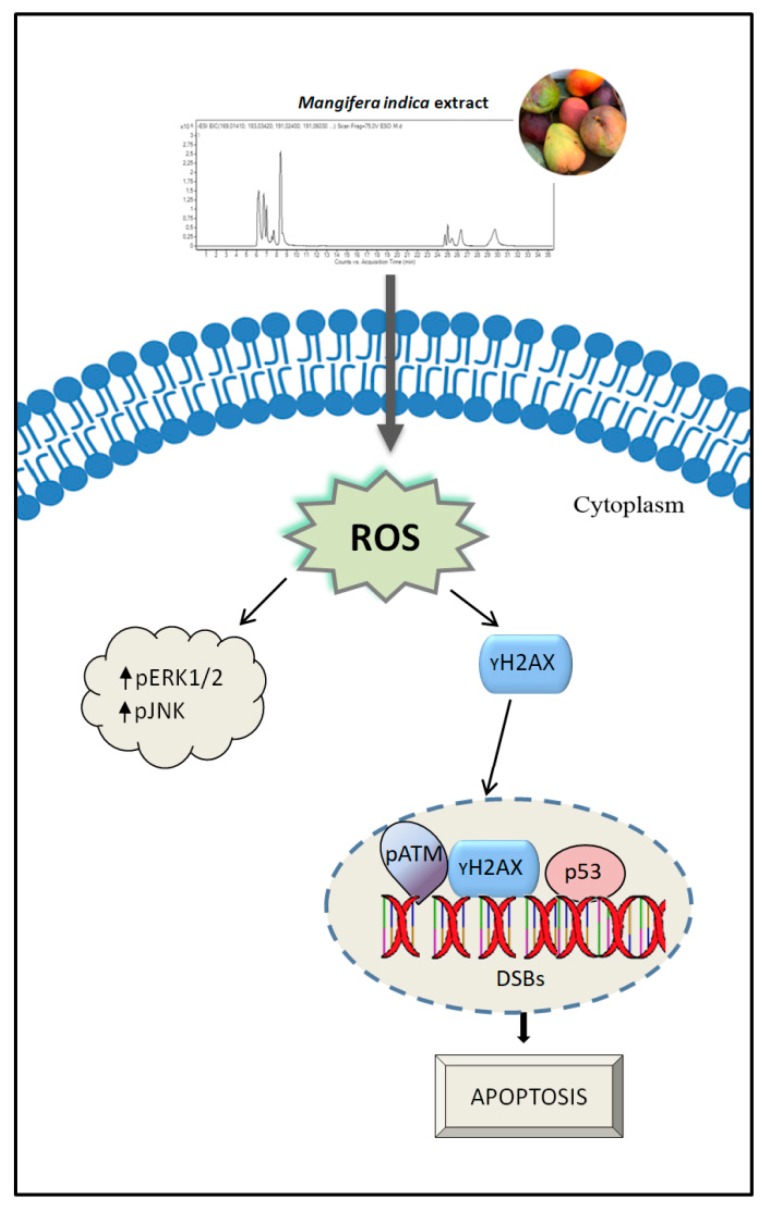
Schematic representation of the mechanism of action of MPE on colon cancer cells. MPE treatment promotes oxidative stress injury evidenced by ROS generation and activation of stress-mediated responses (pERK1/2 and pJNK) as well as DNA fragmentation. The activation of γH2AX in response to DNA damage causes the commitment to apoptotic demise.

**Table 1 antioxidants-08-00422-t001:** Composition of mango peel extract (MPE).

	Compound	RT (min)	ESI^−^ [M − H]^−^ (*m*/*z*)		Molecular Formula	mg/100 g
			*Teor.*	*Exp.*		
**1**	Disaccaride	6.22	341.1089 [M − H]^−^ 377.0856 [M + Cl]^−^ 387.1144 [M + FA]^−^	341.1090 [M − H]^−^ 377.0857 [M + Cl]^−^ 387.1145 [M + FA]^−^	C_12_H_22_O_11_	ND ^a^
**2**	Quinic acid	6.70	191.0561 [M − H]^−^	191.0562 [M − H]^−^	C_7_H_12_O_6_	ND ^a^
**3**	Glucosyl gallate	7.00	331.0671 [M − H]^−^	331.0673 [M − H]^−^	C_13_H_16_O_10_	108.14
**4**	Gluconolactone	7.53	223.0459 [M + FA]^−^	223.0460 [M + FA]^−^	C_6_H_10_O_6_	ND ^a^
**5**	Lepidimoic Acid	7.70	965.2627 [3M − H]^−^	965.2614 [3M − H]^−^	C_36_H_54_O_30_	ND ^a^
**6**	Citric acid	8.39	191.0197 [M − H]^−^	191.0198 [M − H]^−^	C_6_H_8_O_7_	ND ^a^
**7**	Gallic acid	8.63	169.0142 [M − H]^−^	169.0141 [M − H]^−^	C_7_H_6_O_5_	118.57
**8**	Maclurin mono-O-galloyl-glucoside	24.74	575.1042 [M − H]^−^	575.1041 [M − H]^−^	C_26_H_24_O_15_	72.03
**9**	Mangiferin	25.01	421.0776 [M − H]^−^	421.0776 [M − H]^−^	C_19_H_18_O_11_	2.81
**10**	Maclurin di-O-galloyl glucoside	25.34	727.1152 [M − H]^−^	727.1145 [M − H]^−^	C_33_H_28_O_19_	20.21
**11**	Digallic acid	25.46	321.0252 [M − H]^−^	321.0255 [M − H]^−^	C_14_H_10_O_9_	Trace
**12**	Maclurin tri-O-galloyl-glucoside	26.21	879.1262 [M − H]^−^	879.1253 [M − H]^−^	C_40_H_32_O_23_	6.05
**13**	Tetragalloyl glucose	26.31	787.0999 [M − H]^−^	787.0998 [M − H]^−^	C_34_H_28_O_22_	4.88
**14**	Methylgallate	26.36	183.0299 [M − H]^−^	183.0301 [M − H]^−^	C_8_H_8_O_5_	225.87
**15**	Pentagalloyl glucose	26.73	939.1109 [M − H]^−^	939.1100 [M − H]^−^	C_41_H_32_O_26_	17.89
**16**	Methyl-digallate ester	29.68	335.0409 [M − H]^−^	335.0411 [M − H]^−^	C_15_H_12_O_9_	487.15

^a^ Not determined.

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
