# Peer review of "The Anti-Cancer Effect of Mangifera indica L. Peel Extract is Associated to γH2AX-mediated Apoptosis in Colon Cancer Cells"

_antioxidants, 2019, doi:10.3390/antiox8100422_

Round 1

Reviewer 1 Report

The manuscript clearly examines the Anti-cancer Effect of Mangifera indica L. Peel Extract is Associated to H2AX-mediated Apoptosis in Colon Cancer Cells. The paper is interesting and provides new information about the biological activity of Mangifera indica L. The research is design appropriate and the methods clearly explained; the interpretation of the results is clearly presented and adequately supported by the evidence adduced, and the conclusions are logically valid and justified.

Author Response

Thank you so much, we appreciated reviewer’s comments.

Reviewer 2 Report

The paper „The Anti-cancer Effect of Mangifera indica L. Peel  Extract is Associated to gH2AX-mediated Apoptosis  in Colon Cancer Cells” by Lauricella et al. describes  an interesting and comprehensive study in order to explain biochemical mechanisms of anticancer action of mango peel extract.  This work is well-organized, the scientific level meaningful and backed up by a variety of modern experiments. The studies fully fit in the scientific area of Antioxidants journal and seem to be in great interest of wider group of readers, including: biologists, chemists, biochemists, pharmacologists or pharmacognosists.

In my opinion this paper is prepared enough to be published in MDPI Antioxidants journal 

Author Response

(The authors gave the same response as above.)

Reviewer 3 Report

Dear Editor,

I reviewed the manuscript by  Lauricella   et al., entitled ‘’The Anti-cancer Effect of Mangifera indica L. Peel Extract is Associated to H2AX-mediated Apoptosis in Colon Cancer Cells’’

The manuscript is interesting . However, there are a lot of comments on the presented data.  Accordingly, it will be interesting if the authors perform additional experiments by the  treatment of the cell lines with citric acid  and gallic acid to ensure whether the induced effects result from these two products.  Also, the evaluation of induced apoptosis using Annexin V/PI is important. Als, the analysis of reactive oxygen species by flow cytometry will support this study. Also, the quality of the presented pictures is very bad  particularly Western blot. The authors need to replace the Western blot with a better a quality.

Many thanks

Author Response

Comments: The manuscript is interesting . However, there are a lot of comments on the presented data. Accordingly, it will be interesting if the authors perform additional experiments by the treatment of the cell lines with citric acid and gallic acid to ensure whether the induced effects result from these two products. Also, the evaluation of induced apoptosis using Annexin V/PI is important. Als, the analysis of reactive oxygen species by flow cytometry will support this study. Also, the quality of the presented pictures is very bad particularly Western blot. The authors need to replace the Western blot with a better a quality.

Authors’ Reply: Thank you so much, we appreciated reviewer’s comments. In light of the observations, we performed additional experiments to evaluate whether the observed effects in MPE-treated cells can be
ascribed to citric acid and gallic acid. To this purpose, we used a range of concentrations of both gallic acid (GA) and citric acid (CA) to treat all
three colon cancer cells for 48h. Using these compounds at the efficacious doses reported in literature, our studies demonstrated that while GA did not exert any toxic effect, CA markedly reduced cell viability in both Caco-2 and HCT116 cells. In addition, when the cells were exposed to GA/CA combined treatment, no synergist effects were observed.

Since these represent only preliminary data, they were not included in the revised manuscript. Further studies will be performed in the future to explore the concentrations of all components of MPE and analyze the effect of single phytochemicals, used alone or in combination, to identify possible additive or synergistic interactions.

As far as the evaluation of apoptosis, in addition to the analysis of apoptotic markers reported in the manuscript, since we do not have available at the moment the AV/PI kit, we reported data performed by dual
staining with AO/EB. This test, as reported in literature (Liu, X. Med. Sci. Monit. Basic Res., 2015), represents a valuable technical approach to study apoptosis. These new data have been included in the revised version of the manuscript (pages 10-11, lines 318-323, and page 4, lines 167-171 for method description).
As far as the analysis of reactive oxygen species by flow cytometry, we performed an initial investigation on HT29 cells by flow cytometry, demonstrating a consistent ROS generation in the first phase of treatment.

The data obtained induced us to better explore the mechanism of ROS production in all colon cancer cells also evaluating MPE effects also in the presence of ROS scavengers, such as N-acetylcysteine and apocynin,
in time course experiments. Therefore, considering the amount of data reported in the Figure 5 of the manuscript and since the fluorometric analysis is considered a valuable technical approach to demonstrate the
production of ROS, as reported by literature (Wojtala et al., Methods Enzymol. 2014;542:243-62), we chose not to include these data in the new version of the manuscript. Finally, for some western blotting analyses the developed filters were stripped and the detection was repeated and optimized. Therefore images of some western blot were replaced in the revised version (page 14: Nrf2 analysis; page 15: pATM analysis).

Reviewer 4 Report

in this paper  Lauricella and coworkers demonstrate that the anti-cancer effect of Mangifera indica L. peel extract is associated to gammaH2AX-mediated apoptosis in colon cancer cells.

Although this is an interesting study I have some points of concerns:

The best growth inhibitory effect occurs with MPE is observed at 360 ug/ml .I assume that this concentration is very hardly achievable in the in vivo setting. Do the Authors have in vivo anticancer efficacy of MPE in rodent models ? Or do they know or speculate what dose should be administered to humans to achieve these levels in vivo ? Along the same reasoning of 1, the Authors should discuss , and even better prove, whether or not MPE may synergize with standard chemotherapeutic drugs . Any chance of this ? at least preliminary data in vitro ? What was the concentration of LPS of their MPE preparation ? The Authors have convincingly demonstrated that MPE induces ROS production and that this is an important pharmacological mechanism of  chemotherapeutic action. This is interesting. Along the line of studying ROS, the Authors should discuss (or even better show) what is the effect of MPE on production of nitric oxide (NO). NO and NO donors are strong chemotherapeutic agents against blood and solid tumours  and mangiferin has been shown to increase NO release.  If data on NO production are not availaible this issue should be discussed and relevant references quoted and discussed  

Yang H et al.Mangiferin alleviates hypertension induced by hyperuricemia via increasing nitric oxide releases.J Pharmacol Sci. 2018 Jun;137(2):154-161. doi: 10.1016/j.jphs.2018.05.008. Epub 2018 Jun 6.

Xu Y  A switchable NO-releasing nanomedicine for enhanced cancer therapy and inhibition of metastasis.Nanoscale. 2019 Mar 21;11(12):5474-5488. doi: 10.1039/c9nr00732f.

 Paskas S et al., Lopinavir-NO, a nitric oxide-releasing HIV protease inhibitor, suppresses the growth of melanoma cells in vitro and in vivo.Invest New Drugs. 2019 Feb 1. doi: 10.1007/s10637-019-00733-3. [Epub ahead of print]  

Basile MS et al.,Anticancer and Differentiation Properties of the Nitric Oxide Derivative of Lopinavir in Human Glioblastoma Cells.Molecules. 2018 Sep 26;23(10). pii: E2463. doi: 10.3390/molecules23102463.    

Maksimovic-Ivanic D et al., The NO-modified HIV protease inhibitor as a valuable drug for hematological malignancies: Role of p70S6K.Leuk Res. 2015 Oct;39(10):1088-95. doi: 10.1016/j.leukres.2015.06.013. Epub 2015 Jun 28.  

Seabra AB1, Durán N Nitric oxide donors for prostate and bladder cancers: Current state and challenges.Eur J Pharmacol. 2018 May 5;826:158-168. doi: 10.1016/j.ejphar.2018.02.040. Epub 2018 Mar 1.

Author Response

Authors’ Reply:
We thank the reviewer for the insightful comments that we discuss below point by point:
1. The best growth inhibitory effect occurs with MPE is observed at 360 ug/ml .I assume that this concentration is very hardly achievable in the in vivo setting. Do the Authors have in vivo anticancer efficacy of MPE in rodent models ? Or do they know or speculate what dose should be
administered to humans to achieve these levels in vivo?
The anticancer effects of mango peel extracts have not been still demonstrated in vivo against colon cancers. In addition, no data are present in literature to permit to calculate the doses of MPE achieved in humans.
However, a recent study of Parvez et. al. (J Phytopharmacol., 5(3): 112, 2016) demonstrated that the administration of 100 mg/kg daily dose of mango peel extracts to tumor bearing mice is capable of markedly reducing tumor cell growth and increasing life span of mice. Considering these results, we are in
program to explore MPE action in colon carcinoma bearing mice in future investigations.
2. Along the same reasoning of 1, the Authors should discuss , and even better prove, whether or not MPE may synergize with standard chemotherapeutic drugs . Any chance of this ? at least preliminary data in vitro?

In light of this observation, we performed some in vitro experiments combining MPE with doxorubicin (Doxo), an anthracycline drug commonly used in chemotherapeutic protocols. As shown in the figure below, when colon cancer cells were exposed to MPE/Doxo combined treatment no
additive or synergistic effects were found.

However, since such data are only preliminary, they were not included in the revised manuscript, but a sentence to discuss this aspect was reported in the revised manuscript (page 18, lines 534-537). Indeed, we believe that the inspiring reviewer’s advice paves the way to other future fields on
investigation. We aim to test in future studies different combinations between MPE and other chemotherapeutic agents (cisplatin, camptotecin ect.) to find synergistic interactions after calculation of the combination index.

3. . What was the concentration of LPS of their MPE preparation ?
Our analysis did not provide evidence of the presence in MPE of 3-hydroxytetradecanoic acid, that has been reported as indirect evidence of LPS presence (Pais de Barros, J Lipid Res, 2015). However, the direct
determination of LPS will be explored in future investigations, since it could be a possible NO inducer.

4. The Authors have convincingly demonstrated that MPE induces ROS production and that this is an important pharmacological mechanism of chemotherapeutic action. This is interesting. Along the line of studying ROS, the Authors should discuss (or even better show) what is the effect of MPE on production of nitric oxide (NO). NO and NO donors are strong chemotherapeutic agents against blood and solid tumours and mangiferin has been shown to increase NO release. If data on NO production are not availaible this issue should be discussed and relevant references quoted
and discussed.
Another aspect that deserves particular attention is certainly the involvement of NO. As highlighted by the reviewer, our data provided evidence that MPE is able to induce oxidative stress. In accordance with the reviewer’s request, we cannot exclude the possible production of NO in our conditions. Indeed, mangiferin, the bioactive compound present in M. indica L., that we also identified in our HPLC/MS analysis, is able to promote NO release and eNOS generation (Yang H et al., J Pharmacol Sci. 2018). However, beyond the received feedbacks, we discussed these aspects in the text (page 17, lines 481-489) quoting the more relevant references (page 22, references 52-55) accordingly.
Since these aspects deserve particular attention and open the way to a new area of research, we intend to study the role of NO, exploring NO production by Griess test, iNOS and nitrosative stress in future directions, also in light of possible combinations between MPE and NO inducers or NO-donating compounds to enhance the current therapeutic approaches.

Round 2

Reviewer 3 Report

Dear Editor,

I reviewed the Manuscript by Luricella et al .

the manuscript is now improved and can be published in the present form

Many thanks

Reviewer 4 Report

The Authors have adequately addressed my criticisms